# Application of Ni/SiCw Composite Material in MEMS Microspring

**DOI:** 10.3390/mi14091767

**Published:** 2023-09-14

**Authors:** Liyan Lai, Guanliang Yu, Guilian Wang, Yigui Li, Guifu Ding, Zhuoqing Yang

**Affiliations:** 1School of Science, Shanghai Institute of Technology, Shanghai 201418, China; 2School of Electronic and Electrical Engineering, Shanghai University of Engineering Science, Shanghai 201620, China; 3National Key Laboratory of Advanced Micro and Nano Manufacture Technology, Shanghai Jiao Tong University (SJTU), Shanghai 200240, China

**Keywords:** MEMS devices, Ni/SiCw, W microspring, mechanical properties

## Abstract

The microspring is a typical type of device in MEMS devices, with a wide range of application scenarios and demands, among which a popular one is the microelectroformed nickel-based planar microspring prepared by the UV-LIGA technology based on the SU-8 adhesive. It is worth noting that the yield strength of the electrodeposited nickel microstructure is low, and the toughness of the structure is not high, which is unbeneficial for the enduring and stable operation of the spring. The paper mainly presents the methods of preparing high-aspect-ratio Ni/SiCw microstructures for MEMS devices based on UV-LIGA technology, developing Ni/SiCw-based microspring samples with a thickness of 300 μm, and applying a DMA tensile tester for mechanical property tests and characterization. In addition, the paper explores the influence of heat treatment at 300 °C and 600 °C on the tensile properties and microstructure of composite coatings. The results show that the W-form microspring prepared from Ni/SiCw composites not only has a wider linear range (about 1.2 times wider) than that of pure nickel material but also has a stronger resistance capacity to plastic deformation, which is competent for MEMS device applications in environments below 300 °C. The research provides a frame of reference and guidance for improving the stable cyclic operating life of such flat springs.

## 1. Introduction

The microelectromechanical system (MEMS) is an important technology field that has been successfully developed based on microelectronics technology in recent decades, which integrates mechanical and electrical properties into a very small-scale device and can be applied to prepare microdevices ranging from micrometers to millimeters. Highlighted by sensors and actuators, numerous MEMS devices have been widely integrated into the human world, profoundly changing our work and lifestyle. For example, the prosperous development of smartphones and portable healthcare devices that tightly utilize MEMS devices has strongly demonstrated the high market demand for MEMS devices.

The microspring is a typical type of device in MEMS devices, with a wide range of application scenarios and demands, and is an important component of MEMS safety systems, microaccelerometers, microgyroscopes, and inertial switches, not only providing them with elastic force but also transmitting energy [1,2,3,4,5,6]. Therefore, the performance of microsprings is of great significance and decides whether the devices they formed can meet design requirements and function normally. Microsprings manufactured by MEMS micromachining technologies, including Ultra Violet Lithography, Galvano-formung, Abformung (UV-LIGA), Lithographie, Galvano-formung, Abformung (LIGA), and Deep Reactive Ion Etching (DRIE), primarily apply planar structures, such as the S-form and the W-form, among which the W-form is a representative planar microspring. To meet the performance requirements of this device, many scholars conducted numerous simulation calculations and testing verification on the relationship between its structural parameters and deformation characteristics. Lishchynska and Fukushige et al. studied the influence of structure and form parameters of the straight beam and conical microspring, respectively, on device performance under electrostatic force [7,8] and proposed to use chip testing structures to test the mechanical properties of the designed springs, as well as to adopt impact, static, and dynamic loading methods to obtain chronic fatigue characteristics. Li Xiaojie [9] proposed the formula for the vertical elastic coefficient of the “S-type” planar microspring and concluded that the vertical elastic coefficient of the spring decreased with the increased number of nodes, bending radius, and length of the straight beam but increased with the increased width and thickness of the beam. For nickel-based W-form planar microspring structures manufactured by UV-LIGA technology, Qi Xinglin [10] studied the elastic-plastic deformation characteristics of the nickel-based W-form spring and analyzed the influence of the structural parameters of the W-form spring on its deformation characteristics, with the support of elastic-plastic mechanics theory. He Guang [11,12] studied the rigidity characteristics of the W-type microspring, verified the derived microspring constant formula through actual tensile experiments of nickel microsprings, and obtained the internal relationship between rigidity and structural parameters. Li Hua [13,14,15,16] designed and manufactured an “L-form” MEMS microspring with LIGA manufacturing technology, derived its elastic coefficient formula, and demonstrated the accuracy of the formula through experiments. However, influenced by the spring stress concentration and the equivalent pure bending moment at the bending angle, more efforts are desired to improve the accuracy of the formula. Wu Zhiliang et al. [17] optimized the elastic coefficient formula with ANSYS finite element simulation software. Through the experiment verification, the data show that the average error of the elastic coefficient formula for the optimized microspring is lower than 1%, achieving high accuracy. Walraven [18] mentioned that when the stress exceeded the strength limit of the material, the micromechanical structure would be fractured, or the yield would be failed. In addition, structural components would fatigue under a certain period of alternating stress, resulting in the fracture. Man [19] reviewed some general methods for addressing the reliability and qualification of MEMS devices in space applications. The reference mentioned that improving surface smoothness and reducing contact area were effective methods to reduce wear, but the mechanical properties of MEMS devices based on silicon materials were incapable of resisting wear in the long term [20,21].

With the continuous development of MEMS technology, non-silicon materials are increasingly applied in MEMS. Compared with silicon-based MEMS technology, there are many types of non-silicon MEMS materials, including metals, ceramics, glass, polymers and diamonds. Many structural/functional materials have been used to a certain degree due to their attractive advantages in different types of non-silicon MEMS devices and systems, such as high temperature resistance, corrosion resistance, wear resistance, impact resistance, and lightweight high strength, biocompatible, etc. Therefore, a large number of unique high-performance MEMS devices were born based on these materials, showing great development potential and broad application prospects of non-silicon MEMS. However, in general, the maturity of non silicon MEMS is less than that of silicon-based MEMS. Therefore, it is very important to strengthen the application research of non-silicon material preparation and micro machining technology to promote the development of non-silicon MEMS technology.

Currently, the most frequently discussed content in the references is the microelectroformed nickel-based planar microspring prepared by the UV-LIGA technology based on SU-8 adhesive [22,23,24]. It is worth noting that the yield strength of electrodeposited nickel microstructure is low, and the toughness of the structure is not high, which is unbeneficial for the enduring and stable operation of the spring. Usually, they are applied in situations requiring a high service life, such as MEMS safety systems and inertial switches. Certainly, if scientific research can make breakthroughs in material modification, its application range will be greatly expanded. The combination of nickel with one-dimensional nanomaterials is considered to be one of the most effective ways to improve the mechanical properties of nickel-based materials. Silicon carbide whiskers (SiCw) consist of a single crystal with a very ordered structure, which has strong tensile strength, making it a good reinforcement for composite materials, and they have received widespread attention [25,26,27,28,29]. A large number of studies have shown that SiCw reinforced metal matrix composites have the advantages of high specific modulus, high tensile strength, good wear resistance, and dimensional stability. Based on these advantages, SiCw was originally simply used to strengthen the metal matrix [30], but now it has been widely used in electronic devices such as sensors, field emission diodes, and solar cells [31]. In summary, SiC whiskers (SiCw) are very effective nickel-based composite materials for the reinforcing phase. However, whisker-reinforced metal matrix composites are mainly prepared by conventional hot pressing [32], squeeze casting [33], and spark plasma sintering [34]. It is difficult for these traditional powder metallurgy and other preparation processes to be compatible with MEMS processes.

Applying the structural parameters of W-form microspring that former scholars optimized and designed (such as the length, minimum line width, and thickness of the W-form microspring) and replacing the conventional electrodeposited nickel with high-strength Ni/SiCw composite materials, this study aimed to increase the mechanical properties of microsprings and provide a frame of reference and guidance for improving the stable cyclic operating life for such flat springs. Combining the successfully developed silicon carbide whisker reinforced nickel (Ni/SiCw) composite material preparation technology [35,36] in the early stage and integrating with the UV-LIGA technology flow, the study focused on the integrated manufacturing of high-aspect-ratio Ni/SiCw-based microsprings and tested and characterized their mechanical properties, as well as observed and analyzed the cross-sectional morphology of tensile testing. In addition, the influence of heat treatment at 300 °C and 600 °C on the tensile properties and microstructure of the composite coating is comprehensively discussed.

## 2. Experimental and Measuring Procedure

### 2.1. Materials and W-Form Spring Specimen Fabrication

Commercial pure Ti plates with 76.2 mm diameter and 1 mm thickness were prepared. The type of SU-8 resist 1075 (SU-8 from MicroChem Corporation, Newton, MA, USA) was used in this study. The selected commercial SiCw (β-SiCw, XFNANO Material Co., Ltd., Nanjing, China), with an average length of 50 μm and a diameter of 0.2 μm, was added to the bath and maintained in suspension by continuous magnetic stirring with a rate of 300 rpm for at least 12 h before codeposition. After electrodeposition, the W-form spring specimens were ultrasonically cleaned in distilled water for 10 min to remove loosely adsorbed SiCw from the surface. Table 1 lists the composition of the electrolyte and the operating parameters for the electrodeposition. Analytical reagents and distilled water were used to prepare the plating solution.

Combining the structural parameters of the W-form microspring that previous scholars [10,11,12] optimized and designed, the authors of this study drew a mask plate for the microspring, with the basic unit structure in Figure 1, where d is the spacing between beams, b is the line width of the microspring, L is the length of the unit beam, θ is the angle between the curved beams (0 < θ < π), R1 is the inner arc radius of the bending beam in the middle of the spring, R2 is the inner arc radius of the bending beam at the end of the spring (R2 < d/2), L1 is the end connected straight beam (L1 = (b + d)ctg(θ/2)), and L2 is the straight beam at the end of the spring (L2 = d − 2R2), with the corresponding structural parameters as in Table 2.

In our earlier work [37], the bonding strength reinforced substrate and photoetching processes optimized in earlier stages are applied to prepare high-aspect-ratio SU-8 adhesive molds. Combining the UV-LIGA process based on the SU-8 adhesive and Ni/SiCw composite material preparation technology successfully developed in the earlier stage during electrodeposition, a complete technology for Ni SiCw composite material high-aspect-ratio microstructure formation was developed, with the key process flow in Figure 2 and the contents as follows: (1)Preparing high-purity titanium sheets, sequentially processing by polishing with coarse to fine sandpaper, cleaning with deionized water, washing with ethanol, soaking in NaOH for oil removal, cleaning with deionized water, and drying for backup;(2)Anodic oxidation of titanium sheets;(3)Spin-coating SU-8 1075 photoresist in a thickness of 300 μm and standing for 15 min (note: before applying the adhesive, please plasma etch the titanium sheet substrate for 10 min, and then heat treat the substrate at 150 °C for 30 min);(4)Pre-baking: heating rate 2 °C/min, 95 °C insulation for 2 h (note: before baking, please level the hot plate first);(5)Exposing: with a dose of 1.52 J/cm^2^;(6)Baking after exposing: heating rate 2 °C/min, 95 °C insulation for 1.5 h;(7)Development: 20 min;(8)Baking after development: heating rate 2 °C/min, 110 °C insulation for 1 h;(9)Electroformed Ni/SiCw-based W microspring;(10)Sequentially processing the microspring by polishing with coarse to fine sandpaper, cleaning with deionized water, and drying;(11)Releasing: obtaining the Ni/SiCw-based W microspring device.

### 2.2. Tensile Test Procedure

The Dynamic Thermomechanical Analyzer (DMA-Q800, TA Company, Boston, MA, USA) device was applied to test the mechanical properties of the prepared Ni/SiCw composite W-type microspring, and two ends of the microspring were clamped on the movable and fixed ends of the device, with a partially enlarged view of the DMA stretching equipment and clamping device shown in Figure 3. The testing environment was at room temperature, with the testing parameter settings of the DMA strain rate mode, strong ramp, 0.001 N of preload force, and 0.1%/min of strong rate. The surface morphology and microstructure of MEMS springs were characterized by using scanning electron microscopy (SEM; ULTRA55, Zeiss, Germany).

## 3. Results and Discussion

### 3.1. Preparation and Application of High-Aspect-Ratio SU-8 Mold

Through multiple integration experiments of the technology process, the study optimized the implementation effect of the photoetching molding technology and obtained high-quality photoresist molds, as shown in Figure 4a,b. Referring to the SEM image of the SU-8 adhesive mold shown in the figure, it was clear that the boundary of the SU-8 microstructure was neat, the structure was complete, the side walls were steep and smooth, and almost no defects could be observed, which strongly proved that the molding effect of the photoresist mold was prominent and highly capable of meeting the application requirements of high-precision microelectroforming of metal microstructures.

As shown in Figure 5, the Ni/SiCw composite W-form microspring prepared by the UV-LIGA manufacturing technology was sequentially polished with coarse sandpaper to fine sandpaper, cleaned, and dried. The device was mainly composed of eight basic units, with a total length, thickness, and width of the microspring of 2.8, 0.3, and 1.1 mm, respectively. Referring to the information in Figure 5, the side walls of the microspring devices prepared with high-quality photoresist molds were also very steep and smooth, and the structure was complete.

### 3.2. Mechanical Properties of Ni/SiCw-Based Microspring

As shown in Figure 6, the microsprings prepared from pure nickel and Ni/SiCw composites were subjected to tensile testing, and stress–strain curves were obtained. The image on the right shows an enlarged stress–strain diagram of a pure nickel-based microspring. Referring to the information in the figure, when the strain was about 10%, the pure nickel-based microspring had already entered the plastic deformation stage, while the Ni/SiCw-based microspring was still in the elastic deformation stage. Compared with the pure nickel-based microspring, its elastic stage increased by 1.2 times (with a strain of 22%). Compared with the nickel-based W-type microspring manufactured by the UV-LIGA technology reported in Reference [10], its elastic stage increased by 1.1 times, showing that the Ni/SiCw-based microspring was less likely to suffer from plastic deformation. The results indicated that the Ni/SiCw composite material had higher performance and was more competent for application in MEMS devices with higher material strength than that of the pure nickel material.

To explore the function and mechanism of the Ni/SiCw composite microspring, we characterized and analyzed the fracture surface of the Ni/SiCw microspring by using SEM, and the results are shown in Figure 7. As can be seen from the figure, the port of the Ni/SiCw microspring shows obvious sliding characteristics at room temperature [38]. Influenced by the mutual restraint between nickel grains, the embedded SiCw ran through the nickel grains, which played a strong role in hindering the dislocation of nickel grains, so that the crystal slip did not move on a single slip surface but crossed each other and formed a serpentine slip pattern with ups and downs [39,40]. In addition, the whiskers were pulled out from the matrix, and some holes were left, which showed that they overcame the resistance of the nickel matrix and the interfacial shear resistance during the tensile test. Moreover, the tensile section was rather uneven, the typical dimple structure could be seen, and the direction in which the dimple was stretched lacked regularity, which showed that the reinforcing base SiCw deflected the crack path of the coating during the tensile process. At the same time, the dimple structure showed that the Ni/SiCw microspring still had the toughness characteristics of pure nickel.

### 3.3. Influence of Heat Treatment on the Mechanical Properties of Ni/SiCw-Based Microspring

To study the applicable temperature for the W-type microspring prepared in this study, the microspring prepared from the Ni/SiCw composites was subjected to heat treatment. Considering that the conventional usage environment of nickel-based MEMS devices was generally lower than 200 °C, 300 °C was initially selected for the microspring heat treatment for 6 h. Meanwhile, to verify the ultimate temperature that the spring would withstand, the microsprings prepared in the same batch were subjected to a 600 °C heat treatment for 6 h. As shown in Figure 8, the tensile test curves of the microsprings prepared from the Ni/SiCw composites were obtained before and after the heat treatment. In accordance with the test results and after the heat treatment at 300 °C, the mechanical properties of the W-type microspring slightly decreased compared with those before heat treatment, with a relatively insignificant change, but after heat treatment at 600 °C, the mechanical properties of the W-type microspring significantly decreased, the linear range of its tensile curve shortened, and the slope of the stress–strain curve also significantly decreased, showing that its rigidity decreased, as well as its yield range.

To investigate and analyze the reason why the mechanical properties decreased after the heat treatment, we conducted SEM observations on the tensile section of the spring after heat treatment. The SEM image of the fracture morphology after 6 h of tensile testing at 300 °C and 600 °C heat treatment is shown in Figure 9. By comparing Figure 7 and Figure 9, we know that the nickel grains after the heat treatment were much larger than those under the room temperature conditions, and with the temperature increases, the grains gradually increased. When the temperature was 600 °C, the grain size significantly increased, which was because nickel grains underwent recrystallization after the high-temperature pretreatment. In accordance with Formula (1) [41,42], we can further know that with the increased grain size, the yield strength weakened, and the plasticity of the coating decreased. The smaller grain size was capable of improving the stress concentration and reducing the generation of cracks. Therefore, the mechanical properties of the coating at room temperature were more prominent than those of the high-temperature pretreatment at 600 °C.
H = H0 + kd^−1/2^(1)
where k is the constant, known as the Hall–Petch stress intensity factor; d is the size of the microcrystal; H is the yield stress of the material at 0.2% deformation, usually replaced by microhardness; and H0 is the lattice frictional resistance generated when moving a single dislocation.

In addition, influenced by the mismatch between the thermal expansion coefficients of the SiCw and nickel material, the thermal stress at the Ni/SiCw interface increased, causing interface separation between Ni and SiCw. The generated thermal stress after cooling caused these micropores to grow and aggregate to larger pores, where the microspring was prone to fracture at the defect location, thereby reducing its mechanical properties. The results showed that the microspring prepared from the Ni/SiCw composites was applicable for MEMS devices in environments below 300 °C but was not competent for higher temperatures, especially for environments up to 600 °C, which would cause the coating to be fractured at the defect location.

## 4. Conclusions

The innovation of this study is the use of SiC whisker-reinforced nickel-based composite material instead of pure nickel for MEMS microsprings. We mainly focused on and studied how to prepare a high-aspect-ratio Ni-SiCw-based MEMS microspring via the UV-LIGA technology process.

Combining the Ni-SiCw composite electrodeposition method, the paper developed the high-aspect-ratio Ni-SiCw-based microspring samples and tested their tensile properties. The specific conclusions are as follows:
(1)Through testing the tensile performance of the microspring, the results showed that the elastic stage of the Ni/SiCw-based microspring increased by 1.2 times that of the pure nickel-based microspring, with stronger resistance to plastic deformation.(2)Through conducting heat treatment on the microspring at different temperatures, the results showed that the microspring prepared from the Ni/SiCw composites was competent for application to MEMS devices in environments below 300 °C.


In view of the above-mentioned method and results, this paper can serve as the basis for further optimizing the plastic deformation resistance of springs to meet the needs of MEMS.

## Figures and Tables

**Figure 1 micromachines-14-01767-f001:**
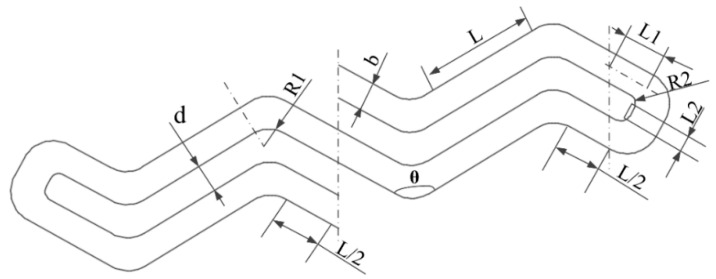
The fundamental element structure of the W-form spring.

**Figure 2 micromachines-14-01767-f002:**
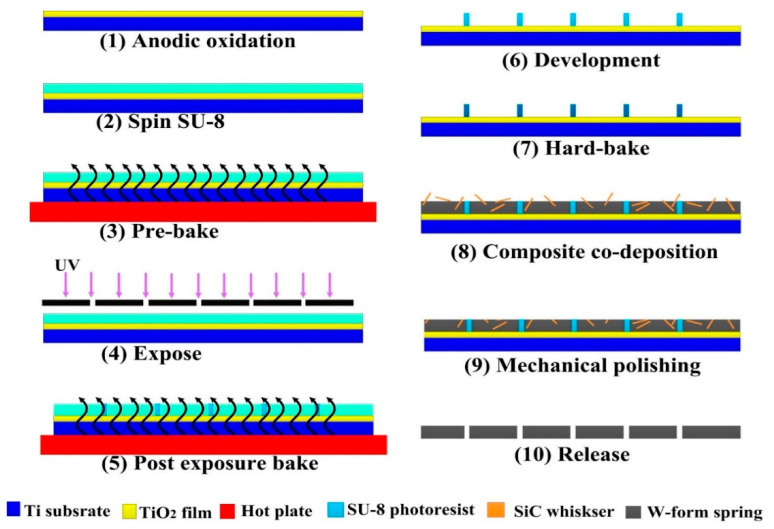
Processing steps for the fabrication of the MEMS spring.

**Figure 3 micromachines-14-01767-f003:**
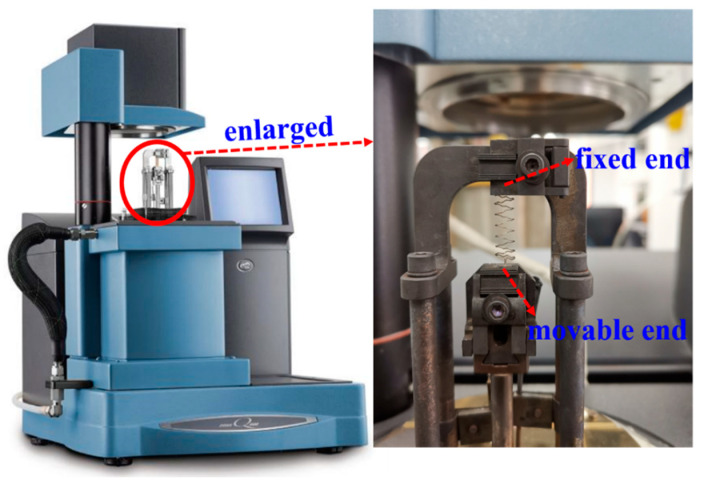
DMA device (**left**) and the partially enlarged photo of its clamping device (**right**).

**Figure 4 micromachines-14-01767-f004:**
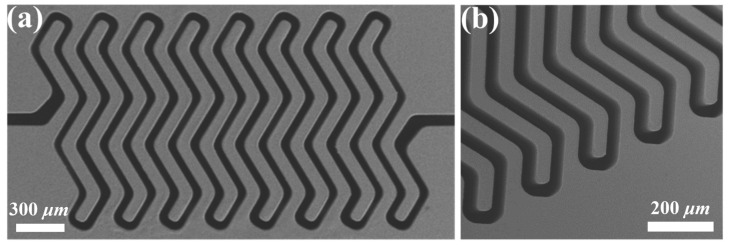
(**a**) SU-8 mold for preparation of the W-form spring and (**b**) the enlarged image of (**a**).

**Figure 5 micromachines-14-01767-f005:**
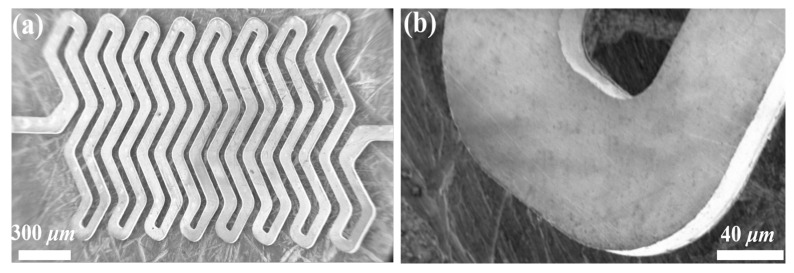
(**a**) The SEM images of the W-form spring electrodeposited by the SiC whisker-reinforced nickel matrix and (**b**) the enlarged image of (**a**).

**Figure 6 micromachines-14-01767-f006:**
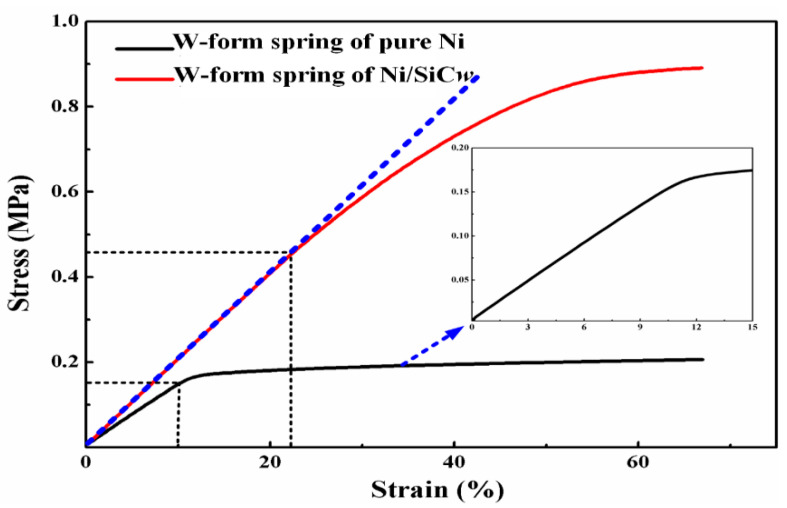
The stress–strain curve of the W-form spring.

**Figure 7 micromachines-14-01767-f007:**
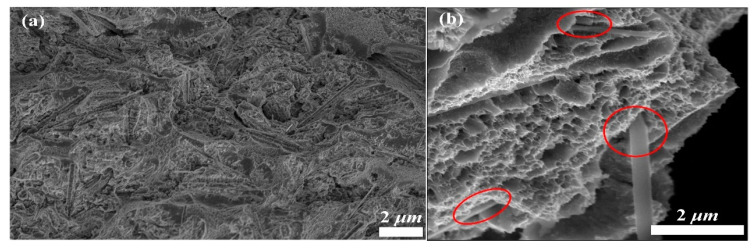
SEM images of the fracture surface of the W-form spring: (**b**) is the enlarged image of (**a**).

**Figure 8 micromachines-14-01767-f008:**
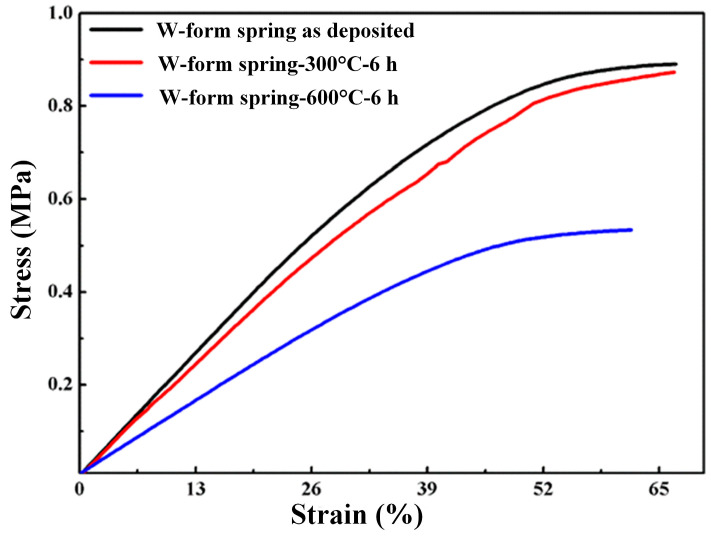
The stress–strain curves of the W-form spring before and after heat treatment at different temperatures for 6 h.

**Figure 9 micromachines-14-01767-f009:**
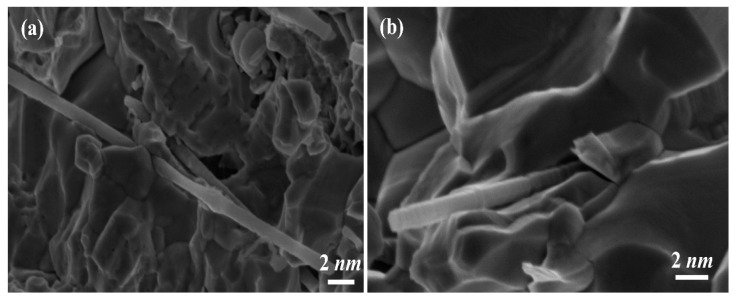
(**a**) The stress–strain curves of the W-form spring before and after heat treatment at different temperatures for 6 h. (**b**) SEM image of W-form spring annealing at 600 °C.

**Table 1 micromachines-14-01767-t001:** The bath composition and electrodeposition parameters.

Deposition Parameter	Amount
Ni (NH_2_SO_3_)_2_·4H_2_O	300 g/L
NiCl_2_·6H_2_O	40 g/L
H_3_BO_3_	30 g/L
C_12_H_25_O_4_SNa	0.03
SiC whisker	1.0 g/L
Applied current	DC
Current density	18 mA/cm^2^
Temperature	43 ± 2 °C
pH	4.1
Stirring speed	300 rpm

**Table 2 micromachines-14-01767-t002:** The structural parameters of the W-form microspring.

Parameter	Numerical Value
b/μm	55
d/μm	55
L/μm	211.5
R1/μm	50
R2/μm	20
θ	2π/3
n	8

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
