# Peer review of "Application of Ni/SiCw Composite Material in MEMS Microspring"

_micromachines, 2023, doi:10.3390/mi14091767_

Round 1
Reviewer 1 Report
The paper is about method of the MEMS spring design from Ni/SiCw composites. The paper is well written, the technology is described in details. The obtained springs have very promising characteristics. It can be interesting for researchers in field of MEMS devices and technology.
The paper can be published in the present form.
Author Response
We thank the referee very much for the careful review and for the positive feedback.
Reviewer 2 Report
The aim of this manuscript is to present a method for fabricating microstructure springs for MEMS devices. The manuscript is well-written and requires only minor revisions based on the following comments:
· The references are outdated and should be updated with more recent ones that reflect the current state of the art in this field.
· The quality of Fig. 8 is poor and should be enhanced to make the details more visible and clear.
· The conclusion section is weak and should be improved by summarizing the main findings, highlighting the novelty and significance of the work, and suggesting possible directions for future research.
· The abstract can be improved by making it more concise and informative.
· The text can be written in a more straightforward way and improved
Author Response
Response to the reviewer #2:
The authors wish to thank the anonymous reviewer for his/her critical comments that helped immensely in improving this manuscript. The changes have been made in this manuscript in red fonts, and the detailed explanations are as follows:
Reviewer #2:
The aim of this manuscript is to present a method for fabricating microstructure springs for MEMS devices. The manuscript is well-written and requires only minor revisions based on the following comments:
- The references are outdated and should be updated with more recent ones that reflect the current state of the art in this field.
Response:
Thank you for your question. The references have been updated with more recent ones in the revised manuscript.
- The quality of Fig. 8 is poor and should be enhanced to make the details more visible and clearer.
Response:
Thank you very much for the careful review. The quality of Fig. 8 has been enhanced in the revised manuscript.
- The conclusion section is weak and should be improved by summarizing the main findings, highlighting the novelty and significance of the work, and suggesting possible directions for future research.
Response:
We appreciate the reviewer’s valuable suggestion. The innovation of this article is to use silicon carbide SiC whisker reinforced nickel matrix composites instead of pure nickel, and this paper can serve as the basis for further optimizing the plastic deformation resistance of springs to meet the needs of MEMS. The related expression has been added in the revised manuscript.
Once again, we thank the referees very much for the careful review and the constructive comments and suggestions.

Reviewer 3 Report
Manuscript ID: micromachines-2566992
Type of manuscript: Article
Title: Application of Ni/SiCw Composites Material in MEMS Microspring
The Authors have reported Ni/SiCw based micro spring for MEMS applications. The manuscript covers the fabrication process of Ni/SiCw micro spring, experimental characterization to estimate the strain behaviour. Further the effect of heat treatment on the strain behaviour of the micro spring. The reviewer feels that the manuscript is not up to the mark regarding the quality of the work and the way the manuscript is written/presented. The authors need to improve the quality of the manuscript and add additional value in terms of the characterization of the micro spring. The reviewer feel that the manuscript can be considered for publication in Micromachines after following major revisions.
1. The reviewer observed that the authors have written the manuscript with lengthy sentences in every section. The authors must consider rewriting the manuscript with better language which can help readers to understand better.
2. In section 1 Introduction, “ranging from millimeters to micrometers” should be changed to micro meters o millimetres.
3. Rectify the grammatical errors thoroughly, for example
In section 1 paragraph 2 “function” to be replaced by “functions”.
In section 1 paragraph 2 “structure calculations” to be replaced by “structural calculation”.
Above are some of the examples, please revisit the manuscript and rectify such errors.
4. Elaborate on UV-LIGA, LIGA, and DRIE, mention the full from of thee process which can help readers to understand better
5. Elaborate on S spring and W spring, how are these different, what are applications, why W spring why not S spring, what are the advances of W springs over S springs etc. Include schematic of both the configurations.
6. What do you mean by “calculation formula” use a better terminology
7. It’s advisable to put flow chart to represent the process flow of the fabrication process.
8. Include a table to show the dimensional specification of the W spring
9. What is the intent of selecting 300 °C and 600 °C for temperature study, what happens to the performance of the W spring beyond 600 °C.?
10. What is the equation 1 used for. No description in manuscript.
11. The author should try to include the fatigue performance of the W spring in the manuscript. Plan appropriate study for the same and include some observations
Overall the manuscript needs thorough editing to improve the quality.
The manuscript requires through revaluation and editing to improve the quality of language and presentation.
Author Response
Response to the reviewer #3:
We thank the referee very much for the careful review and for the positive feedback. We also appreciate the constructive criticism and suggestions that make our manuscript more readable and reasonable.
Reviewer #3 (Remarks):
The manuscript covers the fabrication process of Ni/SiCw micro spring, experimental characterization to estimate the strain behaviour. Further the effect of heat treatment on the strain behaviour of the micro spring. The reviewer feels that the manuscript is not up to the mark regarding the quality of the work and the way the manuscript is written/presented. The authors need to improve the quality of the manuscript and add additional value in terms of the characterization of the micro spring. The reviewer feel that the manuscript can be considered for publication in Micromachines after following major revisions.
- The reviewer observed that the authors have written the manuscript with lengthy sentences in every section. The authors must consider rewriting the manuscript with better language which can help readers to understand better.
Response:
Thank you for the meticulous review. Lengthy sentences have been corrected in the manuscript.
- In section 1 Introduction, “ranging from millimeters to micrometers” should be changed to micro meters o millimetres.
Response:
Thank you very much for the careful review. “ranging from millimeters to micrometers” has been revised as “ranging from micrometers to millimeters”. (Section 1, paragraph 1)
- Rectify the grammatical errors thoroughly, for example, In section 1 paragraph 2 “function” to be replaced by “functions”. In section 1 paragraph 2 “structure calculations” to be replaced by “structural calculation”. Above are some of the examples, please revisit the manuscript and rectify such errors.
Response:
- Thank you for the meticulous review. “function” to be replaced by “functions”, “calculations” to be replaced by “structural calculation”. The others grammatical errors have been revised in the revised manuscript.
- I Elaborate on UV-LIGA, LIGA, and DRIE, mention the full from of thee process which can help readers to understand better.
Response:
Thank you for your comment. The full name of the UV-LIGA, LIGA, and DRIE are 'Ultra Violet Lithography, Galvano-formung, Abformung', 'Lithographie,Galvano-formung, Abformung' and 'Deep Reactive Ion Etching', respectively. They have been added in the revised manuscript. (Section 1, paragraph 2)
- Elaborate on S spring and W spring, how are these different, what are applications, why W spring why not S spring, what are the advances of W springs over S springs etc. Include schematic of both the configurations.
Response:
For microsprings with fixed structural dimensions within a certain range, the optimal spring type is selected based on the elastic coefficient required by device design to fully utilize the limited size space. Under the same external dimensions, line width, thickness, and beam spacing conditions, W-type microsprings can provide a larger flexibility coefficient per unit structural area and occupy a smaller structural area per unit elastic coefficient, but the stiffness of a planar S-shaped microspring is greater than that of a planar W-shaped microspring. These characteristics make W micro springs more suitable for security warning agency. {1. Jianjian Cheng, Weirong Nie, Zhijian Zhou, et al. Study of Planar Micro-spring Performance Comparison [J]. Journal of Mechanical Manufacturing and Automation, 2014, 43(6):4; 2. Guang He, Gengchen Shi. Comparative Study on Stiffness Characterization of Planar S & W-form Micro-Springs Based on MEMS[J]. CHINESE JOURNAL OF SENSORS AND ACTUATORS, 2008, 21(2):288-291}.
Based on the advantages of Ni/SiCw composite material properties previous studies and the performance characteristics required for W microsprings, we have chosen W microsprings as our application research.
- What do you mean by “calculation formula” use a better terminology
Response:
Sorry for the unclear expression. “calculation formula” has been replaced “formula”. (Section 1, paragraph 2)
- It’s advisable to put flow chart to represent the process flow of the fabrication process.
Response:
Thank you for your question. The main processing steps have been shown in figure 2, and specific methods and precautions such as text description.
- Include a table to show the dimensional specification of the W spring
Response:
Thank you very much for pointing out this for us. The dimensional specification of the W spring has been shown in Table 2 and has been added in the revised manuscript.
Table 2 The structural parameters of W-form microspring
|
Parameter |
Numerical value |
|
b/μm |
55 |
|
d/μm |
55 |
|
L/μm |
211.5 |
|
R1/μm |
50 |
|
R2/μm |
20 |
|
θ |
2π/3 |
|
n |
8 |
- What is the intent of selecting 300 °C and 600 °C for temperature study, what happens to the performance of the W spring beyond 600 °C.
Response:
Thank you for your in-depth consideration of this issue. Considering that the conventional usage environment of nickel-based MEMS devices was generally lower than 200°C, 300°C was initially selected for microspring heat treatment for 6 hours. Meanwhile, to verify the ultimate temperature that the spring would withstand, the microspring prepared in the same batch were subjected to a 600°C heat treatment for 6 hours. In accordance with the results, after heat treatment at 300°C, the mechanical properties of the W-type microspring slightly decreased than that before heat treatment. However, after heat treatment at 600°C, the mechanical properties of the W-type microspring significantly decreased, the linear range of its tensile curve shortened, and the slope of the stress-strain curve also significantly decreased, showing that its rigidi-ty decreased, as well as decreased yield range. We speculate that when the heat treatment temperature exceeds 600°C, the performance will further deteriorate and have no practical value. Therefore, no further research has been conducted on the impact of higher heat treatment temperatures on performance. The detailed expression on this in the manuscript as in page 7, paragraph 2.
- What is the equation 1 used for. No description in manuscript.
Response:
Thank you for your question. equation 1 is used for theoretically explain the reason why the plasticity of microspring decreases after heat treatment at 600°C. This is because after heat treatment, the grain size increases and the yield strength decreases, as shown in equation 1.
- The author should try to include the fatigue performance of the W spring in the manuscript. Plan appropriate study for the same and include some observations
Response:
We appreciate the reviewer’s comment. There have been relevant research reports on the fatigue characteristics of W micro springs (Zhao Wang, Research on the Reliability of Typical Fuze MEMS Microspring [D],2012), and this article mainly focuses on the application of composite materials in micro springs. We are still investigating in detail the relation between the grain size of the Ni/SiCw composite coatings and the fatigue performance of the W spring, which will be published in a separate paper in the future.
Once again, we thank the referees very much for the careful review and the constructive comments and suggestions.

Round 2
Reviewer 3 Report
Authors have addressed all the comments from the reviewer. I do not have any further comments. The manuscript can be accepted for in the current form.
Authors have addressed all the comments from the reviewer. I do not have any further comments. The manuscript can be accepted for in the current form.